# Uremic Toxins Affect Erythropoiesis during the Course of Chronic Kidney Disease: A Review

**DOI:** 10.3390/cells9092039

**Published:** 2020-09-06

**Authors:** Eya Hamza, Laurent Metzinger, Valérie Metzinger-Le Meuth

**Affiliations:** 1HEMATIM UR 4666, C.U.R.S, Université de Picardie Jules Verne, CEDEX 1, 80025 Amiens, France; eya.hamza@etud.u-picardie.fr (E.H.); valerie.metzinger@univ-paris13.fr (V.M.-L.M.); 2INSERM UMRS 1148, Laboratory for Vascular Translational Science (LVTS), UFR SMBH, Université Sorbonne Paris Nord, CEDEX, 93017 Bobigny, France

**Keywords:** chronic kidney disease, uremic toxins, anemia, erythropoietin, indoxyl sulfate, iron, erythropoiesis

## Abstract

Chronic kidney disease (CKD) is a global health problem characterized by progressive kidney failure due to uremic toxicity and the complications that arise from it. Anemia consecutive to CKD is one of its most common complications affecting nearly all patients with end-stage renal disease. Anemia is a potential cause of cardiovascular disease, faster deterioration of renal failure and mortality. Erythropoietin (produced by the kidney) and iron (provided from recycled senescent red cells) deficiencies are the main reasons that contribute to CKD-associated anemia. Indeed, accumulation of uremic toxins in blood impairs erythropoietin synthesis, compromising the growth and differentiation of red blood cells in the bone marrow, leading to a subsequent impairment of erythropoiesis. In this review, we mainly focus on the most representative uremic toxins and their effects on the molecular mechanisms underlying anemia of CKD that have been studied so far. Understanding molecular mechanisms leading to anemia due to uremic toxins could lead to the development of new treatments that will specifically target the pathophysiologic processes of anemia consecutive to CKD, such as the newly marketed erythropoiesis-stimulating agents.

## 1. Introduction

Chronic kidney disease (CKD) is a global public health problem characterized by a progressive loss of kidney function through the accumulation of uremic toxins leading to inflammation and endothelial dysfunction [1,2]. The European Uremic Toxin (EUTox) Work Group has compiled an authoritative list of 146 compounds as known uremic toxins and classified them into 3 groups based on their physicochemical characteristics such as the molecular weight, protein-binding capacity and removal pattern by dialysis [3,4,5,6,7]. Twenty-five solutes (27.8%) extracted from the sum of these uremic compounds are protein bound and are difficult to remove by dialysis, due to their protein binding capacity [8]. Furthermore, uremic toxins interact negatively with biologic functions when they are not eliminated by the kidneys. This interaction will be described more exhaustively in a further part of the manuscript.

Instead of being filtered and excreted by kidney, those uremic toxins accumulate in blood and lead to a deficient erythropoietin (EPO) production by the kidney interstitial fibroblasts (peritubular cells of kidneys), resulting in a defective erythropoiesis due to a decrease in erythrocyte production [9,10]. EPO deficiency and iron deficiency are the main reasons contributing to anemia, a well-known consequence of CKD. In this disease, the best available indicator of kidney function is the glomerular filtration rate (GFR), which estimates the total amount of blood filtered by the kidney that passes through the glomeruli per unit of time. Patients with CKD are defined with a GFR inferior to 60 mL/min/1.73 m^2^ in the span of at least 3 months while the normal GFR of a healthy person is more than 90 mL/min/1.73 m^2^ [11,12,13]. End-stage renal disease (ESRD) occurs when the kidney function is below 15% of its normal capacity. Some of the risk factors that lead to common complications for CKD are shown in Figure 1 [14,15,16].

Among them, anemia, heart disease, mineral and bone disease (e.g., osteoporosis and uremic vascular calcification), high potassium (hyperkalemia) and fluid buildup (when extra fluid from the blood are not removed by the kidneys) constitute common complications of ESRD [17]. In some cases, the complications could be the consequence of a strategy treatment of a risk factor. It is the case of the development of hyperkalemia that is common in patients with heart and kidney disease. Because of the increasing use of drugs that antagonize the renin-angiotensin-aldosterone system (RAAS), hyperkalemia is encountered frequently in patients with established cardiovascular diseases (CVD) [18].

In this review, we mainly focus on CKD-associated anemia as it affects nearly all patients with ESRD [19]. Anemia is defined by the World Health Organization (WHO) as a hemoglobin concentration inferior to 13 g/dL in men and less than 12 g/dL in women [20]. Numerous studies have reported that a high percentage, approximately half of patients with ESRD in the United States, suffer from anemia due to EPO deficiency [11,21,22]. As shown in Figure 2, accumulation of uremic toxins impairs renal EPO synthesis, compromising the growth and differentiation of red blood cells (RBCs) in the bone marrow.

This in turn results in a decrease of RBCs production leading to CKD–associated anemia. This exposes patients to a high-risk for CVD (patients with hypertension and heart failure), which are the main cause of death for those patients [23,24]. Among them, we note the development and progression of a heart failure and of a left ventricular hypertrophy (LVH) that is commonly associated with CKD and anemia [25] (Figure 2).

The most common form of anemia classically seen in CKD is typically normocytic, normochromic, and hypoproliferative. Actually, anemia in CKD is a multifactorial condition, and aside from EPO deficiency, iron is also recognized as a major contributor in this disease [26]. Understanding erythropoiesis and its relationship with EPO and iron, leads to understand in part the pathophysiologic processes of anemia consecutive to CKD.

In this review, we discuss current knowledge of the uremic toxin’s effects, focusing on the molecular mechanisms underlying anemia of CKD that are not completely understood and evaluating the new treatments designed to control anemia.

## 2. Uremic Toxins Are Central Players of the CKD Clinical Course

Uremia is attributed to the retention of several solutes, called uremic solutes or uremic toxins. When they are not removed by the kidneys, uremic toxins interact negatively with biologic functions. There is an evident association between increased concentration of uremic solutes and mortality or morbidity in the epidemiologic studies of CKD patients [27,28,29,30,31]. A first work in 2003 by the EuTox Work Group compiled an exhaustive list of uremic retention solutes, containing their mean normal concentration, their mean/median uremic concentration, their highest uremic concentration reported in patients, and their molecular weight (Table 1) [32].

In 2012, an update of uremic retention solutes by the same group extended the classification to new uremic toxins in continuation of the work published in 2003 [6,32]. Based on their protein binding property and molecular weight, they are classified intro three groups: small water-soluble molecules (<500 kDa), middle molecules (≥500 kDa), and protein-bound solutes. This collaborative group focused on measuring the normal and pathologic serum concentrations.

Compared to the small water-soluble molecules, which are easily removed during a standard dialysis, the middle molecules are removed through dialyzer membranes with a larger pore size [38,39]. Protein-bound molecules, on the other hand, are not easily removed with the current dialytic procedure due to the resistance induced by protein binding [40,41]. Even increasing pore size does not improve the removal [41].

### 2.1. Representative Uremic Toxins: Indoxyl Sulfate and P-Cresyl Sulfate

Of the large group of protein-bound solutes, indoxyl sulfate (IS) and P-cresyl sulfate (PCS), poorly removed by conventional dialysis, are the most studied and representative toxins [42,43,44,45,46].

Indoxyl sulfate is a metabolite derived from tryptophan via indole by colon microbes [47]. Briefly, indole is produced by bacterial degradation of tryptophan that is conjugated by the liver and converted into indoxyl sulfate. PCS is generated from the metabolism of tyrosine and phenylalanine by the intestinal flora and sulfated in the liver [47,48]. Both IS and PCS are small molecules, which are bound to plasma proteins at the extant of 90% (e.g., albumin [49]). Importantly, most of the studies have investigated IS and PCS at the same time. It has been shown that PCS is positively related with IS and its increase is correlated with renal function decline in CKD patients [50].

The following section will thus focus on IS, as it is known to be associated with numerous toxic effects, rather than the whole list of toxins. We summarize in Table 2 the most relevant pathophysiologic roles of IS as found in the literature.

It is now well documented that IS contributes to the progression of kidney failure [34,46,50,51,52,53,54]. Experimental work on uremic rats has demonstrated that administration of either indole (precursor of IS) or IS administration stimulates the progression of glomerular sclerosis [34,51]. Also, IS could be a valuable indicator in estimating the progression of renal function deterioration of renal function in advanced CKD patients [53,54]. Another study demonstrated that serum IS level increased gradually as renal function declined in CKD patients [50]. Furthermore, kidney fibrosis could be induced by IS by activating the intrarenal RAAS [52]. In CKD patients, IS-induced renal toxicity mainly occurs following induction of oxidative stress in endothelial cells due to an increase of NADPH oxidase activity and a decrease in antioxidant defense mechanisms [55]. In addition, IS decreases endothelial cell proliferation and wound repair in an in vitro model of human umbilical vein endothelial cells (HUVECs) [2]. Another in vitro study has shown that IS increases endothelial microparticles (EMPs) release and suggests that measuring EMPs levels could be a new marker of endothelial dysfunction in uremic patients [56]. Moreover, IS is also positively correlated with circulating tissue factor (TF) levels induced by the aryl hydrocarbon receptor (AhR) activation in endothelial cells and peripheral blood mononuclear cells (PBMCs) [57]. The increased TF production could participate in the acceleration of atherogenesis in CKD patients, but the authors suggest that this hypothesis has to be confirmed by relevant clinical studies [57]. Also, another study suggests that IS could modify the efflux transporter P-glycoprotein (P-gp) known to be involved in the nonrenal clearance of drugs through the AhR pathway [58]. Identifying the altered transporters may improve the use of drugs associated with CKD for these patients.

To summarize, IS is associated with the progressive deterioration of renal function, the development of uremic symptoms, alongside the pathogenesis of atherosclerosis, CVD, vascular damage, and mortality [34,46,50,51,52,53,54,59,60,61,62,63,64,65].

### 2.2. IS Effects in Renal Anemia

As described above, the toxic effects of IS are numerous. In this review, we will focus on the pathophysiological roles of IS that lead to the induction of renal anemia. Many of the IS effects in renal anemia are related to an impairment of renal EPO synthesis due to a suppression of transcription of the EPO gene in a hypoxia-inducted factor (HIF)-dependent manner, aggravating the hypoxia on the kidney [43] (discussed later in this review). Although renal anemia is predominantly caused by an impaired EPO production, several other factors (e.g., iron deficiency) can worsen erythropoiesis in renal patients. Furthermore, IS contributes to the suicide of erythrocyte, known as eryptosis, characterized by erythrocyte shrinkage due to extracellular Ca^2+^ entry [66].

Another possible mechanism, described in a study by Adelibieke et al., to explain IS-induced anemia is the suppression of the EPO receptor (EPOR)-AKT pathway (intracellular signal transduction pathway that promotes survival and growth in response to extracellular signals) [67]. They speculated that IS suppresses EPO-induced phosphorylation of EPOR in HUVECs. The same study focuses on an additional mechanism that EPO induces the production of Thrombospondin-1 (TSP-1), an erythroid-stimulating factor. TSP-1 is known to stimulate erythroblast proliferation under EPO [68,69]. This is interesting as IS has been described to diminish EPO-induced TSP-1 expression in HUVECs by suppressing AKT phosphorylation [67].

Asai et al. provided in a recent study a putative molecular mechanism that could cause the progression of renal anemia. They provide evidence that AhR activation plays an important rule on the suppressive effect of IS on HIF activation in HepG2 cells. When AhR is inactivated with a pharmacological AhR antagonist or an AhR-siRNA, they observe an abolishment of the suppressive effect of IS on HIF activation [70]. Moreover, IS might increase phosphatidylserine (PS) exposure and microparticles (MPs) release of RBCs due to the increase of cytosolic (Ca^2+^) [71]. PS is a component of the cell membrane found preferentially in the inner leaflet. It is externalized in the outer membrane in combination with MPs release during apoptosis, providing early detection of apoptotic cells [72]. In the same study, the authors demonstrate that the Procoagulant Activity (PCA), promoting the coagulation of blood, is due to the PS exposure and MPs release that might explain the cardiovascular events in CKD [71].

Another report demonstrated that IS is negatively and significantly associated with the EPO expression in CKD patients [73]. In the same study, using animal models, the inhibitory role of IS in the EPO expression has been indirectly confirmed. In fact, when they treated the CKD rats with AST-120 (oral administered intestinal sorbent that can absorb IS), they observed a decrease in seric IS levels and an increase in EPO expression. In contrast, an observational multicentric study including more patients did not show any link between IS and anemia [74]. A study of a larger cohort of patients suffering from CKD is needed in the future to determine whether or not uremic toxins play an important role in renal anemia.

A recent study, comparing healthy controls (CON-RBC) and hemodialyzed patients (HD-RBC) respectively, investigated the mechanism by which IS induces oxidative stress and eryptosis in RBCs [75]. In this paper, Dias et al. suggests that IS increases reactive oxygen species (ROS) generation and eryptosis in CON-RBC. This happens through the organic anion transporter 2 (OAT2) (present in the RBCs surface) and in both NADPH oxidase activity-dependent and glutathione (GSH)-independent mechanism. These findings lend support to an important role of IS in the development of renal anemia.

To conclude, these findings globally support the role of IS in the regulation and development of renal anemia and are summarized in Table 3.

### 2.3. Other Uremic Toxins Implicated in Erythropoiesis

Apart from IS, some other protein-bound uremic toxins have been reported to play a role in renal anemia process by mechanisms independent of EPO production.

In patients with ESRD, polyamines have been reported to reduce proliferation and maturation of erythroid precursor cells (CFU-E) by acting as inhibitors of erythropoiesis [76]. Some studies about polyamines confirm the hypothesis that polyamines are intimately involved in the development and progression of anemia pathogenesis in ESRD [76,77,78]. For example, acrolein stimulates the formation of ceramide that in turn induces eryptosis by triggering cell membrane scrambling and cell shrinkage when exposed to cytosolic Ca^2+^ [79]. All those studies suggest an important role, not only of IS but also of other uremic toxins, in the development of renal anemia. Further studies will however have to be undertaken to understand an eventual cumulative mechanism of these toxins.

In the field, it is also possible to cultivate cell cultures with serum from uremic patients to see the effect of the sum of uremic toxins accumulated in the blood of CKD patients. To our knowledge, only one study published in 1983 suggests a potential mechanism of uremic sera on erythropoiesis inhibition [80]. This study has shown an inhibitory effect of uremic serum from CKD patients on the proliferation in vitro of erythroid progenitor cells. To date, however, the toxic and inhibitory effects of uremic toxins on erythropoiesis are not totally understood.

## 3. Physiopathology of Anemia in CKD

HIF is activated in order to stimulate the production of EPO in response to a decreased oxygen delivery [81]. In a pathological context, this is the case during the course of anemia. HIF is a heterodimer with a hypoxia-inducible factor alpha-subunit (HIF-α), and a constitutively expressed beta-subunit (HIF-β). The two isoforms of HIF-α (HIF-1α and HIF-2α) are strictly dependent on hypoxia. Moreover, it has been demonstrated that HIF-2α is the most important isoform in mediating the response of anemia as a key regulator of erythropoiesis and iron metabolism [82,83,84,85,86,87].

### 3.1. Regulation of HIF in Kidney Disease

The expression of HIF-α is mainly regulated by proteosomal degradation [88,89] (Figure 3).

Under normoxic conditions, HIF-α is hydroxylated by prolyl hydroxylase domain (PHD) protein [90]. The proline-hydroxylated HIF-1α is then recognized by von Hippel-Lindau (VHL)-E3-ubiquitin ligase complex, polyubiquitinated and degraded in the proteasome. In contrast, under hypoxic conditions, HIF-α escapes from hydroxylation mediated by PHD [90]. HIF-α forms a heterodimer with HIF-β. Therefore, the HIF heterodimer binds to the hypoxia response element (HRE) in the regulatory region of target genes (e.g., EPO) and promotes transcription of these genes.

A study of Bernhardt et al. suggests that PHD inhibitors (FG-2216) can stabilize HIF levels by preventing hydroxylation, promoting dimerization and stimulating endogenous EPO production in the kidneys of hemodialyzed patients, suggesting a new therapeutic approach for CKD [91]. In other words, pharmacologic PHD inhibitors could offer a new approach for the treatment of anemia (discussed later in the review).

Under normoxia, HIF-α is hydroxylated by PHD protein, then recognized by VHL-E3-ubiquitin ligase complex, and directed to the proteasome for degradation. Under hypoxia, HIF-α accumulates in the nucleus and dimerizes with HIF-β, and together they lead to transcription of EPO. Under CKD, the mechanism is similar to the hypoxic condition except that IS activates the AhR receptor that in turn inhibits HIF leading to a decrease of EPO production. 

### 3.2. Impairment of IS-Induced HIF Activation

In CKD, both HIF-1 and HIF-2 are activated, so it is difficult to determine their individual effects on anemia [92]. However, HIF-2α is a key regulator of hypoxic EPO induction [93] and could induce a severe anemia when deleted from renal tissue [94]. As revealed in the study of Chiang et al., IS potentially suppresses EPO expression in a HIF-dependent manner both in vitro and in vivo [43]. In hypoxic condition, IS is able to impair erythropoiesis by suppressing the EPO gene transcription via inhibition of HIF activation. These relationships between HIF and IS were confirmed in an independent study in which IS inhibits the hypoxic induction of HIF-1 target genes causing functional impairment of the HIF-1α C-terminal transactivation domain (CTAD) in human kidney (HK-2) proximal tubular cells [95]. Furthermore, as previously seen, IS activates the transcription factor AhR in HUVECs [57]. In addition, AhR activation plays an indispensable role on the suppressive effect of IS on HIF activation and subsequent EPO production in hepatocellular cells (HepG2) [70]. That could be one of the molecular mechanisms by which renal anemia is caused.

### 3.3. EPO and Regulation of Erythropoeisis

Erythropoiesis is a multistep process in which mature RBCs are generated from hematopoietic stem cells (HSCs) in bone marrow (BM). It is a highly regulated process in which HSCs proliferate and differentiate consecutively into burst-forming unit-erythroid (BFU-E), colony-forming unit-erythroid (CFU-E), proerythroblasts (Pro-EB), erythroblasts (EB), reticulocytes (Retic), and mature RBCs (or erythrocytes) in the circulation, consecutively (Figure 4).

It is known that a defect in erythropoiesis could lead to hematological diseases [96].

EPO is the key hormone that regulates erythropoiesis. In the bone marrow, erythroid cells expressing EPOR are found mainly at later stages, erythroid progenitor CFU-E and proerythroblasts [97]. Furthermore, EPO acts synergistically with other cytokines (e.g., SCF) in the bone marrow to cause maturation and proliferation from the stage CFU-E and Pro-EB (both EPO-dependent) to the erythroblast stage of erythropoiesis [98,99].

The interaction between EPO and its receptor is the main trigger involved in erythropoiesis regulation. The binding of EPO to EPOR stimulates erythroid cell division and proliferation and inhibits erythroid progenitor apoptosis [97]. Upon binding to EPO, EPOR homodimerizes at the cell surface and the signal transduction cascade is initiated [100,101]. An intracellular signaling occurs by the induction of tyrosine phosphorylation of JAK2 kinase, that in turn phosphorylates other proteins to transduce a growth signal [102].

EPO binds to the receptor and generates intracellular signals to rescue the late CFU-E cells from apoptosis and increase their survival and proliferation [99]. During this process, the transcription factor HIF-2α promotes erythropoiesis, by regulating EPO synthesis and enhancing iron uptake [103].

### 3.4. The Role of Iron in the Regulation of Erythropoeisis

Iron is an essential mineral required for hemoglobin synthesis in maturing erythroblasts. Although the progenitors CFU-E and Pro-EB are EPO-dependent, the differentiation process from proerythroblasts to red blood cells (or erythrocytes) is strongly iron-dependent (Figure 4) [104]. Erythropoiesis requires 25–30 mg of iron to produce almost 200 billion red blood cells in the bone marrow each day. This iron is provided primarily from recycling hemoglobin-derived iron of old and senescent red cells. Indeed, less than <1–2 mg of new iron daily derives daily from intestinal absorption [105]. A study revealed that eryptosis could be associated with iron deficiency [106].

### 3.5. Hepcidin and Regulation of Serum Iron Levels

The main hormone that regulates iron levels is hepcidin, a peptide hormone produced in the liver [107]. In turn, hepcidin production is homeostatically modulated by iron concentrations and thereby iron availability for erythropoiesis [108]. During erythropoiesis, hepcidin production is suppressed, thus increasing iron availability for hemoglobin synthesis [109]. In CKD patient’s blood, hepcidin levels are elevated, in part due to inadequate kidney clearance [110]. Therefore, abnormal elevated hepcidin production may contribute to the development of anemia in CKD [111]. There are three identified erythroid modulators of hepcidin expression in ineffective erythropoiesis: Erythroferrone (ERFE), Growth differentiation factor 15 (GDF15), and Twisted gastrulation protein homolog 1 (TWSG1) [105]. In this review, we decided to focus specifically on erythroferrone because it is closely related to EPO [112].

The most recently discovered regulator of hepcidin expression is ERFE, a member of the C1q-tumor necrosis factor-related family of proteins [113]. ERFE is an EPO-responsive gene produced by erythroblasts in response to EPO (Figure 5). During stress erythropoiesis, defined as the rapid production of new erythrocytes, ERFE suppresses hepcidin expression to respond to an increased erythropoietic demand [114]. In Kautz et al. study, EPO injections into mice (within 4 h) resulted in an increase of ERFE mRNA expression by erythroid precursors in the erythropoietic organs (bone marrow and spleen) [113]. In response to anemia, hypoxia induces an increase of EPO production by the kidney, which stimulates erythroblasts to increase ERFE production and thereby suppress hepcidin [115].

As previously shown, IS accumulation suppresses EPO production by inhibiting HIF [43]. A recent study tried to clarify how IS is involved in iron metabolism in CKD [116]. Their in vitro experiments showed that IS enhances hepcidin expression via both AhR and oxidative stress-mediated pathways. Therefore, IS seems to be involved in iron metabolism by impairing iron utilization and promoting an inadequate erythropoiesis in the CKD context.

Also, it is to be noted that even if uremic toxins such as IS and PCS strongly aggravate anemia in advanced CKD and ESRD, renal anemia in this setting is primarily related to a decreased production of EPO by diseased kidneys (with interstitial fibrosis and reduced renal mass of cells producing EPO with the exception of genetic polycystic disease; cases of enlarged kidneys in diabetic and amyloid nephropathies with deposits of amorphous specific material have also a reduction of EPO-producing cells), together with a lowered set point for EPO production in case of hemorrhage. This decrease in EPO production is associated with true iron deficiency related to blood loss due to uremic enteropathy and hemodialysis treatment aggravated by iron-restricted anemia (linked to high levels of hepcidin).

## 4. Current Strategies of Renal Anemia Treatment

Small water-soluble compounds are easily withdrawn by dialysis while other compounds need more specific removal strategies [40]. Indeed, middle molecules can be better removed by dialysis membranes with a larger pore to allow their passage, but this is not the case for the protein-bound compounds [40]. Therefore, at present, a development of new elimination strategies and innovative pharmacological procedures is necessary to prevent the related complications of CKD (Table 4).

The anemia associated with CKD can be managed effectively by the erythropoiesis-stimulating agents (ESA), based on recombinant human erythropoietin (rHU EPO) such as epoetin alfa, or intravenous iron supplementation [117,118]. Most patients with CKD that respond to ESA administration experience correction of anemia, improving quality of life and maintaining hemoglobin concentrations within the recommended target range [119,120]. However, paradoxically, although correcting anemia, some patients seem to be resistant to such treatments [121]. An increase of adverse effects was noted such as worsening hypertension, left ventricular hypertrophy, progression of kidney disease, and higher mortality rate [118,122,123]. In fact, approximately 10% of patients with anemia due to CKD are hyporesponsive to ESA therapy [120]. For a better ESA response in long-term HD patients, the Kidney Disease Outcomes Quality Initiative (KDOQI) guidelines recommend an iron saturation ratio (ISAT) above 20% and a serum ferritin above 200 ng/mL [124]. When those two predictors are less than the recommended values, ESA becomes hyporesponsive [125].

Optimization of ESA therapy may require iron supplementation for treatment of anemia in patients with CKD [126]. Studies in patients with CKD under dialysis demonstrate that the use of iron supplementation can lower ESA dose requirements and might reduce the risks associated with ESA therapy [127,128]. In fact, an inadequate amounts of iron supply commonly causes an unsuccessful ESA treatment [126]. It is now considered that, for an effective anemia treatment in patient with CKD to be effective, requirement of combined ESA therapy and iron administration is needed [138].

Nevertheless, some patients with CKD do not adequately respond to recommended ESA or iron doses. A potential novel therapeutic approach for anemia treatment consists of the stimulation of endogenous erythropoietin secretion through HIF stabilization using orally active HIF prolyl-hydroxylase domain inhibitors (PHIs) [91,129,130,139]. Among them, Roxadustat (FG-4592) is an oral PHI that stimulates erythropoiesis and regulates iron metabolism in ESRD patients. A study has demonstrated the effect of Roxadustat treatment on anemia in patients undergoing dialysis [140]. In the present trial, they compared Roxadustat therapy with epoetin alfa therapy for 26 weeks. This study showed that Roxadustat led to a greater increase of hemoglobin level than epoetin alfa in the treatment of anemia. A complementary study by the same team compared Roxadustat and placebo for the treatment of anemia on patients with CKD who were not undergoing dialysis for an 18-week period [131]. Over week 8, patients administered with Roxadustat treatment had a hemoglobin level increased compared to those with placebo. During the subsequent 18-week open-label treatment period, Roxadustat maintained hemoglobin levels. PHIs are currently being tested in humans and have shown several advantages compared to the ESA currently available: they are simpler and cheaper to produce, can be orally administered, induce more stimulating intrinsic EPO production and decrease cardiovascular risks [132].

Finally, we will discuss the oral absorbent AST-120, which specifically plays a role in the removal of IS from CKD patients. In fact, AST-120 decreases the serum levels of IS by absorbing indole in the intestines and thereby stimulating its excretion into feces [133,134,135,136]. It was shown that AST-120 slowed the progression of CKD but also the appearance of CVD [46]. Another study has reported that AST-120 administration, when associated with an EPO stimulating agent, is associated with an improvement of hemoglobin levels [137]. This study suggests that AST-120 may improve the EPO production by lowering serum IS levels in an animal model [73]. These findings offer new insights to the development of novel therapeutic strategies to manage anemia in late-stage CKD patients.

## 5. Conclusions

In conclusion, anemia associated with CKD is a multifactorial condition that can be induced by EPO and/or iron deficiency. The knowledge of the uremic toxin’s effects demonstrated by the in vivo and in vitro studies allows us to better understand the current molecular mechanisms. This review underlies the physiopathology of anemia in CKD focusing on a potential mechanism of EPO on iron regulation in uremic condition. Therefore, the understanding of the pathophysiological mechanisms gives an overview of the current strategies of treatment to manage renal anemia.

## Figures and Tables

**Figure 1 cells-09-02039-f001:**
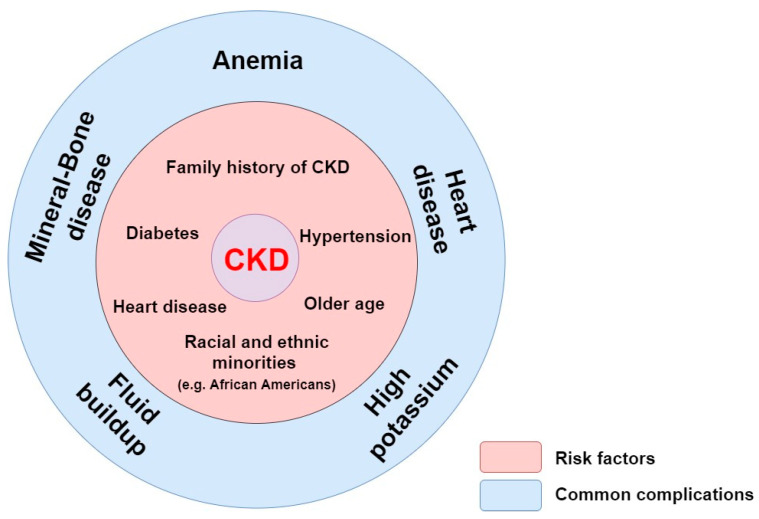
Risk factors and common complications for CKD. In pink are represented the risks factors: family history of CKD, hypertension, diabetes, heart disease, older age, and racial and ethnic minorities. In blue are represented the common complications: anemia, heart disease, mineral and bone disease, fluid buildup, and high potassium.

**Figure 2 cells-09-02039-f002:**
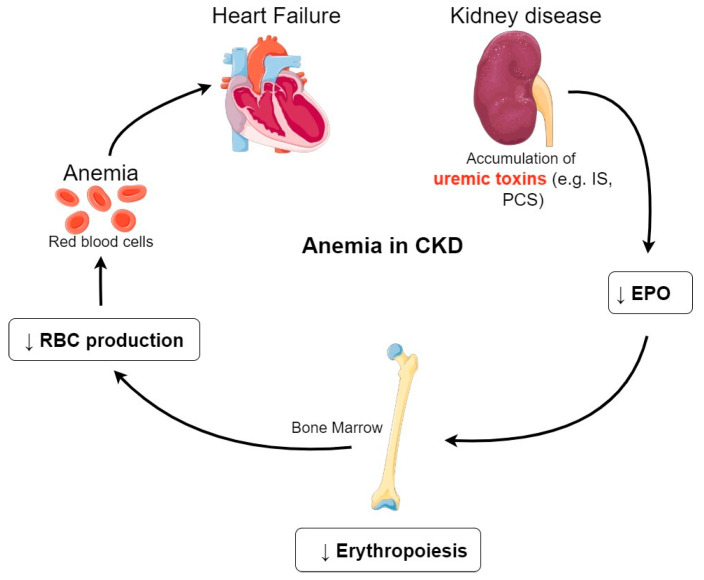
Anemia in CKD. Accumulation of uremic toxins induces a decrease of EPO production in the kidney. The decreased EPO synthesis compromises erythropoeisis in the bone marrow. This in turn results in a decrease of RBCs production leading to CKD-associated anemia. Abbreviations: CKD, chronic kidney disease; EPO, erythropoietin; RBCs, Red Blood Cells; IS, indoxyl sulfate; PCS, P-cresyl sulfate.

**Figure 3 cells-09-02039-f003:**
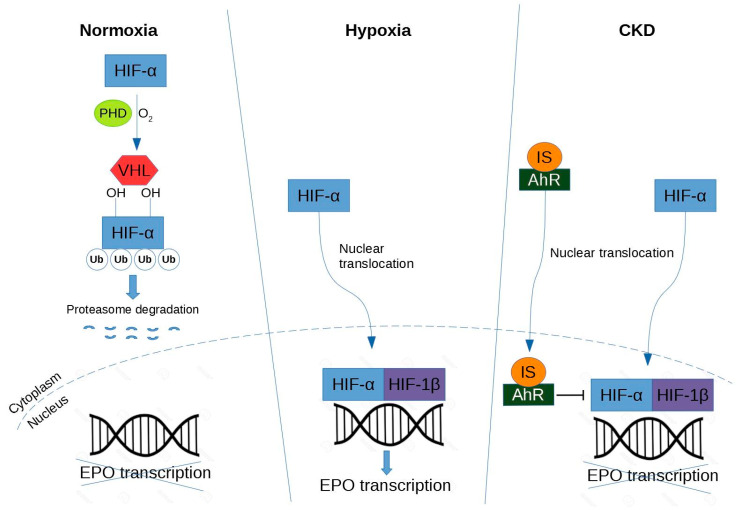
Regulation of hypoxia-inductible factor in renal erythropoietin-producing cells. Abbreviations are: HIF, hypoxia-inducible factor; PHD, prolyl hydroxylase domain; VHL, von Hippel-Lindau-E3-ubiquitin ligase complex; Ub, ubiquitylation; EPO, erythropoietin; AhR, aryl hydrocarbon receptor; IS, indoxyl sulfate.

**Figure 4 cells-09-02039-f004:**
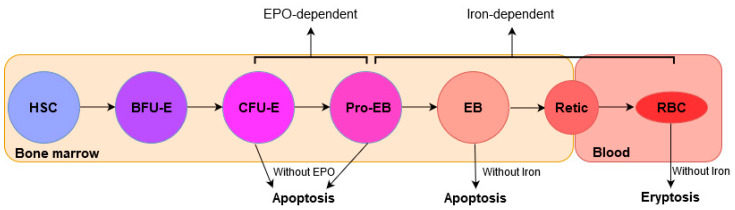
Overview of erythropoiesis. HSC differentiates into RBC, as indicated by the arrows. Stages of erythropoiesis EPO-dependent and Iron-dependent are indicated. Abbreviations are: HSC, hematopoietic stem cell; BFU-E, burst-forming unit-erythroid; CFU-E, colony-forming unit-erythroid; Pro-EB, proerythroblasts; EB, erythroblasts; Retic, reticulocytes; RBC, mature RBCs; EPO, erythropoietin.

**Figure 5 cells-09-02039-f005:**
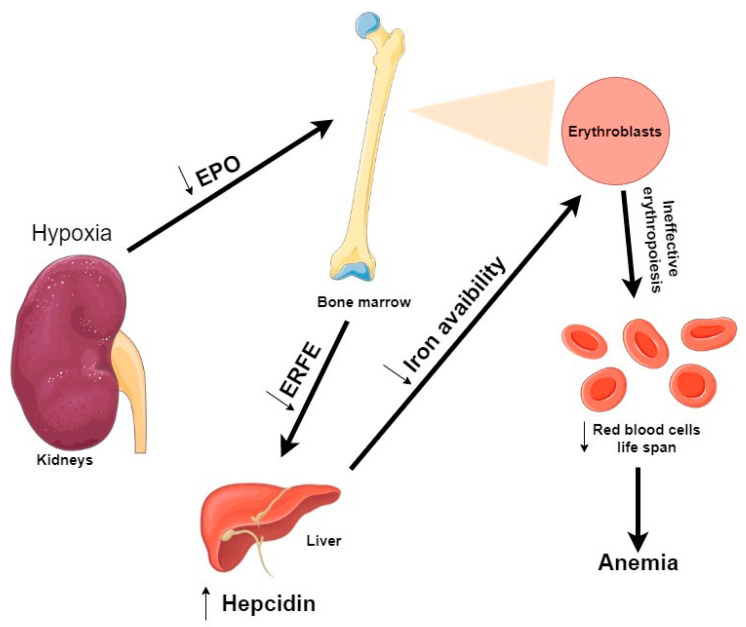
Overview of erythropoiesis regulation in uremic condition. During hypoxia and in a CKD condition, accumulation of uremic toxins by the kidneys leads to a decrease of EPO production. During erythropoiesis, because erythroblasts are less produced, ERFE production is decreased. As ERFE is the regulator of hepcidin, a decrease of ERFE production leads to an increase of hepcidin. Thus, leading to a decrease of iron availability. As a consequence, many erythroblasts incur apoptosis before completing differentiation, causing an ineffective erythropoiesis and fewer red blood cells are produced. Abbreviations: EPO, erythropoietin; ERFE, Erythroferrone.

**Table 1 cells-09-02039-t001:** Physiochemical classification of Uremic Toxins by the EuTox Work Group.

Classification	Representative Solute	C_n_	C_u_	C_max_	MW (kDa)	Ref
Free water-soluble solute	Urea (g/L) Creatinine (mg/L)	<0.4 <12.0	2.3 ± 1.1 136.0 ± 46.0	4.6 240.0	60 113	[32,33]
Protein-bound solute	Indoxyl sulfate (mg/L) P-cresyl sulfate (mg/L)	0.6 ± 5.4 0.6 ± 1.0	53.0 ± 91.5 20.1 ± 10.5	236.0 40.7	251 108	[34,35]
Middle molecule	β-2 microglobulin (mg/L)	<2.0	55.0 ± 7.9	100.0	11818	[36,37]

Abbreviations: C_N_, normal concentration; C_U_, mean/median uremic concentration; C_MAX_, maximal uremic concentration; MW, molecular weight; ref, reference.

**Table 2 cells-09-02039-t002:** The pathophysiological roles of Indoxyl sulfate.

The Pathophysiological Role of Indoxyl Sulfate	Reference
Inhibition of endothelial proliferation and wound repair	[2]
Progressive deterioration of renal function	[34,46,50,51,52,53,54]
Induction of oxidative stress	[55]
Increase of circulating EMPs release	[56]
Induces TF production via the AhR pathway	[57,58]
Development of uremic symptoms	[46,59]
Associated with pathogenesis of atherosclerosis	[60]
Increases mortality	[61]
Cardiovascular disease	[54,61,62,63]
Peripheral arterial disease	[61,64,65]

Abbreviations: EMPs, endothelial microparticles; TF, tissue factor; AhR, aryl hydrocarbon receptor.

**Table 3 cells-09-02039-t003:** Role of IS in the regulation of renal anemia.

The Pathophysiologic Roles of IS	Molecular Mechanisms	References
Impairment of erythropoiesis in a HIF dependent manner	Suppression of the EPO gene transcription during hypoxia	[43]
Stimulates eryptosis	Extracellular Ca^2+^ entry with subsequent stimulation of cell shrinkage and cell membrane scrambling	[66]
Might contribute to EPO resistance and endothelial dysfunction	IS inhibits EPO-Induced Phosphorylation of EPOR IS inhibits TSP-1 expression through suppression of the AKT phosphorylation	[67]
Suppression of HIF activation	IS-induced AhR activation	[70]
Increased PCA in RBCs	Due to PS exposure and RBCs-derived microparticles release	[71]
EPO decrease	IS negatively regulates the EPO expression	[73]
IS-induced RBCs death	Through OAT2, and NADPH oxidase activity-dependent, and a GSH-independent mechanism	[75]

Abbreviations: HIF, hypoxia-inducted factor; EPO, erythropoietin; EPOR, erythropoietin receptor; IS, indoxyl sulfate; TSP-1, Thrombospondin-1; AhR, the aryl hydrocarbon receptor; PCA, Procoagulant Activity; PS, Phosphatidylserine; RBCs, red blood cells; OAT2, Organic Anion Transporter 2; GSH, glutathione.

**Table 4 cells-09-02039-t004:** Current strategies used to treat renal anemia.

Current Strategies	Reference
ESA	[117,118,119,120,121,122,123,124,125]
Iron supplementation	[126,127,128]
PHD inhibitors	[91,129,130,131,132]
AST-120	[73,133,134,135,136,137]

Abbreviations: ESA, erythropoiesis-stimulating agents; PHD, prolyl hydroxylase domain.

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
