# Peer review of "Uremic Toxins Affect Erythropoiesis during the Course of Chronic Kidney Disease: A Review"

_cells, 2020, doi:10.3390/cells9092039_

Round 1
Reviewer 1 Report
In this review, the authors focus on the effects of uremic toxins on the anemia of CKD. The review is well-written, the bibliography is recent and the conclusions are well-reported. The quality of some figures is low, but the article may be suitable for publication after minor corrections.
Author Response
We thank the Reviewers for their positive comment and careful review, which helped improve the manuscript. Please find our answer to comments in blue as well as suggested text changes in the "Track Changes" function in Microsoft Word.
Reviewer 1 :
The quality of some figures is low, but the article may be suitable for publication after minor corrections.
Reply : We improved the figures quality using the correct DPIs.
Reviewer 2 Report
This clearly written and documented review describes in deep the negative role of uremic toxins especially Indoxyl sulfate and P-cresyl sulfate on erythropoiesis during advanced chronic kidney disease (CKD) end end-stage kidney disease (ESKD).
Some points to improve the manuscript :
The first is epistemological : if uremic toxins especially Indoxyl sulfate and P-cresyl sulfate strongly aggravate anemia in advanced CKD and ESKD (as very elegantly shown by the authors), renal anemia in this setting is primarily related to a decrease production of epoetin by diseased kidneys (with interstitial fibrosis and reduced renal mass of cells producing EPO with the exception of genetic polycystic disease; cases of elarged kidneys in diabetic and amyloid nephropathies with deposits of amorphous specific material have also a reduction of EPO-producing cells), together with a lowered set point for erythropoietin production in case of hemorrhage as shown by the literature in the eighties and nineties. This decrease in epoetin production is associated with true iron deficiency related to blood loss due to uremic enteropathy and hemodialysis technic aggravated by iron restricted anemia (linked to high levels of hepcidin).
line 384: KDOQI suggest ferritin above 200 ng/ml (and not below) and TSAT above 20 % (and not below) to reduce the risk of ESA hypo-responsiveness.
Line 396-400: I suggest to the authors to include the analysis of the twin article of he New England Journal of Medicine (Chen et al ; PDF of the article given with the review) performed in hemodialysis patients treated with Roxadustat (and placebo) which strengthens their demonstration .

Author Response
This clearly written and documented review describes in deep the negative role of uremic toxins especially Indoxyl sulfate and P-cresyl sulfate on erythropoiesis during advanced chronic kidney disease (CKD) end end-stage kidney disease (ESKD).
Reply: We thank the reviewer for this kind judgment
Some points to improve the manuscript :
The first is epistemological : if uremic toxins especially Indoxyl sulfate and P-cresyl sulfate strongly aggravate anemia in advanced CKD and ESKD (as very elegantly shown by the authors), renal anemia in this setting is primarily related to a decrease production of epoetin by diseased kidneys (with interstitial fibrosis and reduced renal mass of cells producing EPO with the exception of genetic polycystic disease; cases of elarged kidneys in diabetic and amyloid nephropathies with deposits of amorphous specific material have also a reduction of EPO-producing cells), together with a lowered set point for erythropoietin production in case of hemorrhage as shown by the literature in the eighties and nineties. This decrease in epoetin production is associated with true iron deficiency related to blood loss due to uremic enteropathy and hemodialysis technic aggravated by iron restricted anemia (linked to high levels of hepcidin).
Reply : We completely agree with the reviewer, thank him for this interesting addition to our manuscript and added this paragraph in the end of the third part of the review line 361-368.
line 384: KDOQI suggest ferritin above 200 ng/ml (and not below) and TSAT above 20 % (and not below) to reduce the risk of ESA hypo-responsiveness.
Reply : We corrected this sentence.
Line 396-400: I suggest to the authors to include the analysis of the twin article of the New England Journal of Medicine (Chen et al ; PDF of the article given with the review) performed in hemodialysis patients treated with Roxadustat (and placebo) which strengthens their demonstration .
Reply : We included the analysis of both article of Chen et al. in the text between line 405-415.
Reviewer 3 Report
In the present study the authors review the molecular mechanisms leading to anemia due to uremic toxins. This is an interesting study that may shed some light on the pathophysiology of uremic toxicity. Nevertheless, I have some concerns, please find my comments below.
line 31 - please check, the number of uremic toxins had been updated, they are more than 150 compounds. Please insert a more recent referencePlease improve the figures quality, I imagine that they are not with the correct DPIs.
Table 1 and 2 and all tables – I suggest to city the first author name, instead of reference number as well as a brief description of the study such as – uremic rats, CKD patients cohort, etc…..
Please correct the paragraphs rules in all the manuscript, the text is too fragmented and divided in too many paragraphs.
Author Response
We thank the reviewer for his kind remarks and suggestions.
line 31 - please check, the number of uremic toxins had been updated, they are more than 150 compounds. Please insert a more recent reference Please improve the figures quality, I imagine that they are not with the correct DPIs.
Reply : We changed the number of uremic toxins in line 31 from 90 to 146 (we added the 90 compounds of Vanholder et al. (2013) to the new 56 compounds found in the Duranton et al. (2012) study. It is the only update that we found in the state of art.
Table 1 and 2 and all tables – I suggest to city the first author name, instead of reference number as well as a brief description of the study such as – uremic rats, CKD patients cohort, etc…..
Reply : We couldn’t change the references in the table by citing the first author name instead of the number because it will automatically change all the style for the citation. In this journal the style citation must be IEEE, we couldn’t make the change for only the table.
-Line 183 : we added the model of Asai et al. study “ in HepG2 cells”.
Please correct the paragraphs rules in all the manuscript, the text is too fragmented and divided in too many paragraphs.
Reply : We regrouped some paragraphs in order to improve the manuscript as cleverly asked by the reviewer.